# Potential application values of a marine red yeast, *Rhodosporidiums sphaerocarpum* YLY01, in aquaculture and tail water treatment assessed by the removal of ammonia nitrogen, the inhibition to *Vibrio* spp., and nutrient composition

**Long Yun**[1,2⦿], **Wei Wang**[1,3⦿], **Yingying Li**[1,4], **Mei Xie**[1,4], **Ting Chen**[1,4], **Chaoqun Hu**[1,4], **Peng Luo**[1,4]*, **Daning Li**[1]

1 CAS Key Laboratory of Tropical Marine Bio-resources and Ecology (LMB), Guangdong Provincial Key Laboratory of Applied Marine Biology (LAMB), South China Sea Institute of Oceanology, Chinese Academy of Sciences, Guangzhou, China, 2 Institut Jacques Monod, Université Paris Diderot, CNRS, UMR 7592, Paris, France, 3 Institute of Oceanography, Minjiang University, Fuzhou, Fujian, China, 4 Southern Marine Science and Engineering Guangdong Laboratory, Guangzhou, China

⦿ These authors contributed equally to this work.

* luopeng@scsio.ac.cn

**Data Availability Statement:** The 18S rDNA sequence of the strain was deposited in GenBank under accession number MK583688.

## Abstract

In recent years, marine red yeasts have been increasingly used as feed diets for larviculture of aquatic animals mainly due to their rich nutrition and immunopotentiation, however little attention is given to their other probiotic profits. In this study, a marine red yeast strain YLY01 was isolated and purified from farming water and it was identified as a member of *Rhodosporidiums sphaerocarpum* by the phylogeny based on 18S rDNA sequence. The strain YLY01 could effectively remove ammonia nitrogen from an initial 9.8 mg/L to 1.3 mg/L in 48 h when supplemented with slight yeast extract and glucose in water samples and the removal rate of ammonia nitrogen was up to 86%. Shrimps (*Litopenaeus vannamei)* in experimental group incubated with the yeast YLY01 exhibited a higher survival rate than those in blank control group and positive control group challenged by *Vibrio harveyi*, and it manifested that the strain has high biosecurity to at least shrimps. The strain YLY01 could inhibit the growth of *Vibrio* cells when a small quantity of carbon source was added into farming water. In addition, a nutrition composition assay showed the contents of protein, fatty acids, and total carotenoids of the yeast YLY01 were 30.3%, 3.2%, and 1.2 mg/g of dry cell weight, respectively. All these results indicated that the marine red yeast YLY01 has a great potential to be used as a versatile probiotic in aquaculture and to be developed as a microbial agent for high-ammonia tail water treatment.

**Funding:** This work was supported by Natural Science Foundation of Guangdong Province, China (2019A1515011492), National key Research and development program: Blue Granary Scientific and Technological Innovation (2020YFD0901104), and Key Special Project for Introduced Talents Team of Southern Marine Science and Engineering Guangdong Laboratory (GML2019ZD0402).

**Competing interests:** The authors declare that the research was conducted in the absence of any commercial or financial relationships that could be construed as a potential conflict of interest.

## Introduction

Intensive aquaculture has quickly expanded in recent 30 years as it is an important food resource for a growing global human population and an important way for gaining economic benefits in developing countries [1, 2]. However, the discharge of huge aquaculture wastes and the abuse of toxic chemicals and veterinary medicines have caused food security problems and environmental problems including eutrophication [3], the destruction of natural ecosystem [4], and the dispersal of aquatic pathogens and drug-resistant bacteria [5, 6]. The environmental problems caused by excessive aquaculture without tail water management greatly decreased the success rate of farming in return. Several alternative methods have been considered to improve the quality and sustainability of aquaculture production [7]. Among those methods, probiotics have been shown to have an important role in aquaculture [8].

Yeasts, as one group of probiotics, have been mainly used either as fresh baits for larva of aquatic animals or as feed supplement in aquaculture. Until now, yeast products using in aquaculture are primarily from brewer's yeast and baker's yeast(*Saccharomyces cerevisiae*)[9] and few marine yeast products come into the market mainly because the research on the application of marine yeasts in aquaculture and the product development of marine yeasts are just beginning. Most of previous studies on marine yeasts concentrated on their abilities to enhance the immunocompetence of aquatic animals due to the existence of yeast polysaccharides (such as β-glucans) and to promote the growth of aquatic animals [10, 11], which aroused great interest on further exploring valuable marine yeasts in mariculture.

In present study, a marine red yeast strain first concentrating on the ability of ammonia removal was screened and isolated from farming water. Ammonia and nitrite removal, vibrios inhibition, and nutritional composition were further assayed to assess the potential application value in aquaculture and tail water treatment.

## Materials and methods

### Collection of the water samples

Farming water samples were collected from the bottoms of five different shrimp culture ponds (shrimps have been cultured for nearly three months) in Maoming, Guangdong Province, China, and the collected water samples were stored at 4°C before use.

### Experimental shrimps

Healthy-looking shrimp juveniles, *Litopenaeus vannamei*, were collected from a farming pond of Maoming Jinyang Aquaculture Company in Maoming, Guangdong, China (150 individuals with the body length of 4 ± 1 cm and the body weight of 0.62±0.15 g). The shrimps were further randomly sampled for conventional PCR detection for WSSV [12], EHP [13], and EMS-related vibrios [14] to exclude the infection by these pandemic pathogens. The shrimps were temporarily reared in a 1000-L tank for three days in an experiment base of Maoming Jinyang Aquaculture Company. The collection of water and shrimp samples and experiments were permitted and authorized by Maoming Jinyang Aquaculture Company and South China Sea Institute of Oceanology, Chinese Academy of Sciences, Guangzhou, China.

### Medium preparation and isolation for yeasts

The screening of yeasts was conducted using modified YPD (YPDM) medium containing 10 g of yeast extract, 20 g of peptone, 20 g of glucose, and 1000 mL of filtered seawater (pH was adjust to 7.0). The YPDM medium was sterilized, and supplemented with 1 ‰ (v/v) of

tetracycline (12 mg/mL) and kanamycin (50 mg/mL) before use. Concentrated YPDM (10×, CYPDM) medium (Ph 7.0) was also prepared for enriching yeasts.

Isolation medium for yeasts with the ability of ammonia removal (IMAR) contains 1 g of $(NH_4)_2SO_4$, 5 g of glucose, 5 g of sucrose, 0.5 g of yeast extract, and 0.5 g of $NaH_2PO_4$ in 1000mL of filtered seawater (pH was adjusted to 7.0). Concentrated IMAR (10×, CIMAR) was also prepared to enrich yeasts that have the activity of ammonia removal.

Water samples were mixed with CYPDM medium at a ratio of 9: 1 (v/v) in 250mL conical flasks and then incubated in a shaker (200 rpm) at 30˚C. After 48-h incubation, each of 100-μL cell fluids was spread on fresh YPDM agar plates and incubated overnight at 30˚C. Colonies were picked out based on colony morphology and further purified on YPDM agar plates for at least three times. Five yeast strains were obtained in this way, and then they were inoculated in culture tubes with 4 mL of IMAR for 12 h and 100 μL of each strain was cultured on fresh IMAR agar plates through streaking method and incubated overnight at 30˚C. Single colony was picked into 4 mL of IMAR and incubated in a 200-rpm shaker for 48 h. The initial and the final concentration of ammonia nitrogen in the culture mediums were determined by a μMAC SMART hydraulics rev.3 instrument. The analysis included three replicates. The strain with the most removal of ammonia nitrogen was picked out and used for subsequent experiments.

## Molecular identification of the selected strain

In order to identify the candidate strain, a 18S rDNA-based method was used. The genomic DNA of the strain was extracted using a Yeast Genomic DNA Extraction Kit (TianGen, China) for PCR amplification of 18S rDNA with primers EF3 and EF4 [15]. PCRs were performed as follows: 94˚C for 3 min; 30 cycles at 94˚C for 30 s, 56˚C for 90 s, and 72˚C for 1 min, and a final step of 72˚C for 5 min. The purified PCR product was directly sequenced by Sangon Biotech Company (Shanghai, China). The obtained sequence was queried against the GenBank database using BLASTN (https://blast.ncbi.nlm.nih.gov/Blast.cgi). A phylogenetic tree was constructed by the neighbor-joining (NJ) method [16] based on the sequence of the strain and related sequences using MEGA 6.0 [17] with 1000 bootstrap replicates.

## Observation of cell morphology by scanning electron microscopy

The yeast cells in logarithmic growth phase were collected and centrifuged at 3000 rpm for 5 min, and then the precipitation was suspended with 0.1M PBS (0.14 M NaCl, 3 mM KCl, 8 mM $Na_2HPO_4$, 1.5 mM $KH_2PO_4$, pH 7.4) and repeated for three times. Then the suspension was transferred to a new centrifugal tube and glutaraldehyde was added up to 0.5%. After inoculated at 4˚C for 30 min, glutaraldehyde was added into the sample up to 2.5% at 4˚C overnight. The cells were washed twice with $ddH_2O$ by centrifuging at 3000 rpm for 5 min, followed by the gradient dehydration with 30%, 50%, 70%, 80%, 90%, 95%, 100% ethanol for 20 min at each stage. The dehydrated sample was processed by critical point drying and then was deposited with gold by ion sputtering. The cell morphology was observed and photographed by scanning electron microscopy.

## The effect of the selected strain on ammonia nitrogen removal in simulated farming water

Ammonia nitrogen removal of the selected strain was performed using a method [18] with a slight modification. A total of 1000 mL of the filtered and sterilized seawater (pH 8.0) containing ammonia nitrogen (10 mg/L), carbon sources (1 g/L), and yeast extract (0.5 g/L) was prepared as simulated farming water (SFW). The yeast cells were cultured overnight and adjusted

to a concentration with $OD_{600nm}$ = 1 (approximately $2 \times 10^7$ cfu/mL). Then 3 mL of the cells was centrifuged, washed twice, and resuspended with sterilized seawater. In the experimental group, 500 μL of the suspended cells was added to each of 150-mL SFW, whilst in the control group, 500 μL of sterilized seawater was added to each of 150-mL SFW. The samples were incubated at 30°C in a 150-rpm shaker. The concentration of ammonia nitrogen was measured by the μMAC SMART hydraulics rev.3 instrument at 0, 3, 6, 9, 12, 24, 36, and 48 h. Each of experimental and control group had three replicates.

## The impact of the selected strain on the growth of *Vibrio* Spp.

An experiment was conducted to determine whether the strain can inhibit the growth of *Vibrio* species in farming water [18]. A farming water sample was taken from a shrimp pond suffered by an outbreak of vibriosis, and the number of *Vibrio* cells was calculated by conventional TCBS plates [19]. A total of 2 L of the water sample was added with 2 g of glucose and 1 g of yeast extract. Then, 150 mL of the water sample containing glucose was added into each 250-mL conical flask. Each conical flask in the experimental group was added with 100 μL of washed and resuspended yeast cells ($OD_{600nm}$ = 1.0), whilst each conical flask in the control group was added with 100 μL of sterilized seawater. The experimental group and the control group each contained three replicates. After 48 h of incubation at 30°C in a 150-rpm shaker, the water samples were serially diluted, spread on fresh TCBS plates, and incubated at 30°C overnight to calculate the number of *Vibrio* cells [19].

To explore whether the strain can inhibit the growth of *Vibrio* species through secreting antibacterial substances, we also conducted a plate suppression experiment [18]. *V. alginolyticus* E06333, *V. furnissi* ATCC 33813, *V. harveyi* E385, and *V. parahaemolyticus* ATCC 17802, were used. 100 μL of *Vibrio* cells were spread onto LB plates and then each plate was covered with three 5-mm sterilized filter sheets. The filter sheets were dropped with 5 μL of tested cells [20]. The sizes of the inhibition zones around the filter sheets were observed and measured.

## Bio-safety assessment of the selected strain to shrimp *Litopenaeus vannamei*

Shrimp juveniles, *Litopenaeus vannamei*, were used to assess the biosafety of the selected strain according to a previously reported method [18]. Healthy shrimp juveniles mentioned above were randomly grouped into tanks with 40-L filtered seawater disinfected by chlorine dioxide (South Ranch, China). Experimental group was supplemented with 50 mL of washed and resuspended yeast cells (the final concentration, approximately $1 \times 10^5$ cfu/mL). Blank control group was supplemented with sterilized seawater, and positive control group was supplemented with 50 mL of *V. harveyi* cells (the final concentration, approximately $1 \times 10^5$ cfu/mL). Each group contained three replicates (three tanks). Each tank contained 10 shrimps. During the period of the experiment, the shrimp were fed normally, residual feeds and feces were siphoned quickly, and sterilized seawater (approximately 0.5 L) was complemented each day. The number of survival shrimps was recorded for seven days.

## Nutritional composition analysis of the selected yeast strain

Cell dry weight was calculated by measuring the mass change of 100 mL cell sediment harvested by centrifugation from initial mass to constant weight with drying at 55°C in an oven. For determining the content of carotenoids, 0.1 g of dry cells was weighed and resuspended with 2.4 mL of 3 M HCl, and then the suspension was heated in a boiling water bath for 4 min, followed by rapid cooling and centrifugation for 5 min at 4000 rpm. The cell pellets were washed twice, and then 4 mL of acetone solution was added, followed by vortex for 2 min and

centrifugation for 5 min at 4000 rpm. The supernatant was harvested to measure the content of carotenoids by a BIO-RAD SmartSpec™ Plus spectrophotometer instrument at wave length of 475 nm [21, 22]. Crude protein content was determined by the Kjeldahl method [23], and fatty acids content was determined by the Bligh-Dyer method [24]. Yeast polysaccharide was extracted by pronase digestion and alcohol precipitation and was quantified by a reported method [25]. Amino acid composition of the yeast was analyzed by a Chinese national standard method (GB 5009.124–2016) using an automatic amino acid analyzer.

## Results

### Isolation and identification of the candidate yeast strains

Totally, five strains that could grow in IMAR containing ammonia nitrogen were selected and purified on YPDM agar plates. After 48 h incubation, all of them showed the ability to remove ammonia nitrogen but there were obvious discrepancies in the removal rates of ammonia nitrogen by them (Fig 1). The strain YLY01 showed the strongest ability to remove ammonia nitrogen, and totally 81.9% of ammonia nitrogen in IMAR medium was removed by it in 48 h (from 213.5 mg/L to 38.7 mg/L) (Fig 1), which is 4.8 times removal rate than the strain YLY05 (17.0%). As a result, the strain YLY01 was screened out for the subsequent experiments.

A BLASTN search indicated that 18S rDNA sequence of YLY01 strain had a 99% identity with a *Rhodosporidium sphaerocarpum* strain JCM 8202 (GenBank: AB073275). The results of the phylogenetic tree based on 18S rDNA sequences of the strain YLY01 and other related yeast strains clearly indicated that the strain YLY01 and other strains from *R. sphaerocarpum* were clustered into one branch (Fig 2). Therefore, phylogeny based on 18S rDNA sequences confirmed that the strain YLY01 is one member of *R. sphaerocarpum*. The 18S rDNA sequence of the strain *R. sphaerocarpum* YLY01was deposited in GenBank under the accession number, MK583688.

### Cell morphology of the yeast strain YLY01 under SEM

Similar to most yeast cells [26], YLY01 cells are round or elliptical with an average diameter of 2 μm, and the surface of the cells is smooth and flagellum-free (Fig 3). The germinating and proliferating cells and the bud marks on the surface of the cells could be clearly observed under SEM.

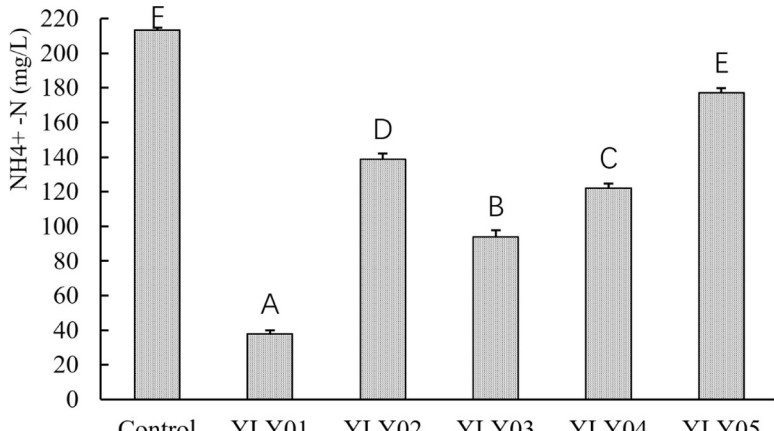

**Fig 1. Ammonia nitrogen removal in IMAR medium by five marine yeast strains after 48-h culture.** Control: blank IMAR medium. YLY01-YLY05: five candidate yeast strains tested. The values at the top of each column represent the concentration of ammonia nitrogen in IMAR medium after 48-h culture. Data are given as mean ± SD (n = 3). Different letters above bars indicate significant difference among treatments (p < 0.05).

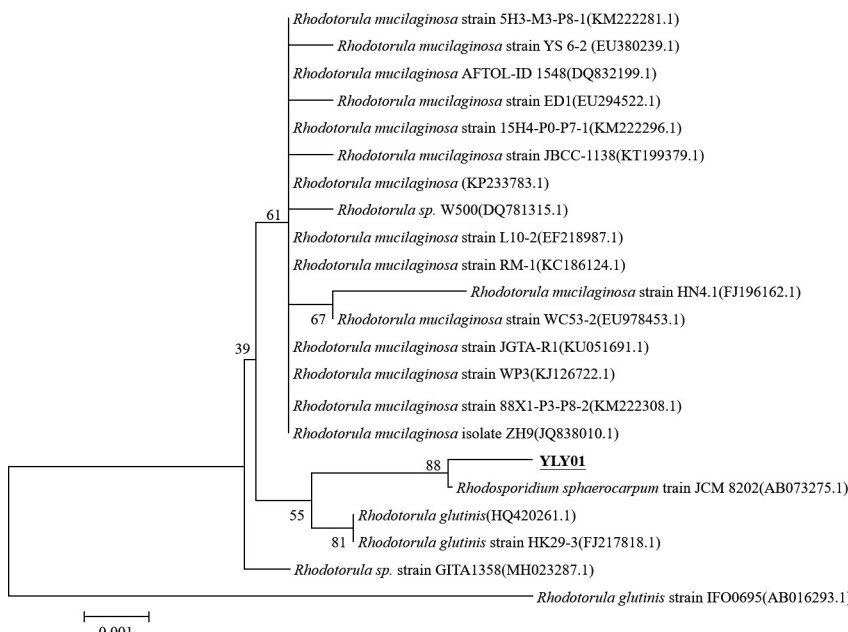

*Rhodotorula mucilaginosa* strain 5H3-M3-P8-1(KM222281.1)
*Rhodotorula mucilaginosa* strain YS 6-2 (EU380239.1)
*Rhodotorula mucilaginosa* AFTOL-ID 1548(DQ832199.1)
*Rhodotorula mucilaginosa* strain ED1(EU294522.1)
*Rhodotorula mucilaginosa* strain 15H4-P0-P7-1(KM222296.1)
*Rhodotorula mucilaginosa* strain JBCC-1138(KT199379.1)
*Rhodotorula mucilaginosa* (KP233783.1)
*Rhodotorula sp.* W500(DQ781315.1)
*Rhodotorula mucilaginosa* strain L10-2(EF218987.1)
*Rhodotorula mucilaginosa* strain RM-1(KC186124.1)
*Rhodotorula mucilaginosa* strain HN4.1(FJ196162.1)
*Rhodotorula mucilaginosa* strain WC53-2(EU978453.1)
*Rhodotorula mucilaginosa* strain JGTA-R1(KU051691.1)
*Rhodotorula mucilaginosa* strain WP3(KJ126722.1)
*Rhodotorula mucilaginosa* strain 88X1-P3-P8-2(KM222308.1)
*Rhodotorula mucilaginosa* isolate ZH9(JQ838010.1)
**YLY01**
*Rhodosporidium sphaerocarpum* train JCM 8202(AB073275.1)
*Rhodotorula glutinis*(HQ420261.1)
*Rhodotorula glutinis* strain HK29-3(FJ217818.1)
*Rhodotorula sp.* strain GITA1358(MH023287.1)
*Rhodotorula glutinis* strain IFO0695(AB016293.1)

0.001

**Fig 2. A phylogenetic tree based on 18S rDNA sequences of the strain YLY01 and related strains.** Bootstrap values were obtained after 1000 repetitions. Scale bar indicates 0.1% sequence dissimilarity.

## Ammonia nitrogen removal in SFW by the yeast strain YLY01

After 48-h treatment with the yeast YLY01 in the experimental group, the ammonia nitrogen concentration in SFW declined from initial 9.8mg/L to 1.3 mg/L, with a removal rate of 86.7%

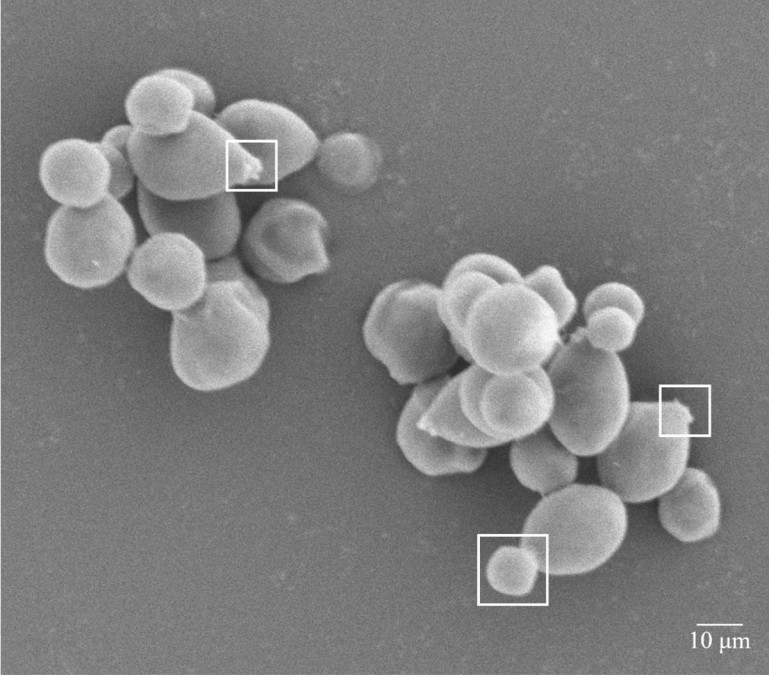

10 μm

**Fig 3. The morphology of the yeast YLY01cells under SEM.** The bud marks (shown in write boxes) on the surface of some cells could be clearly observed.

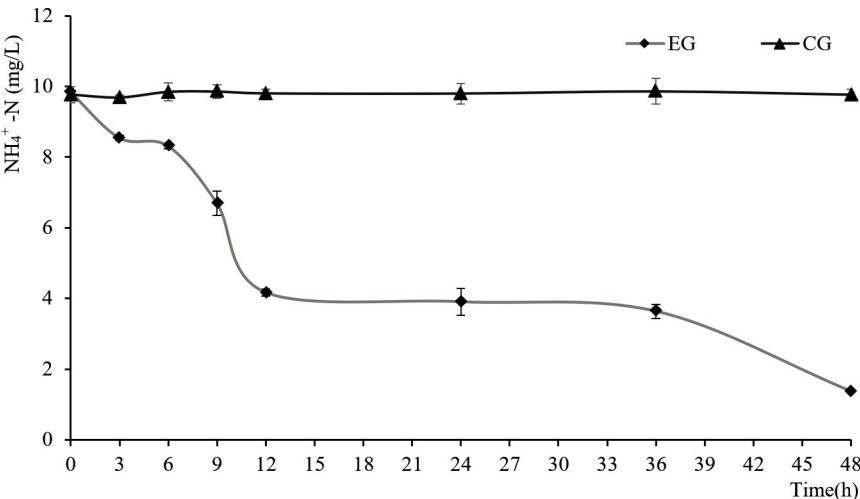

**Fig 4. The changes of ammonia nitrogen content of SFW in 48-h incubation.** EG: The experimental group treated with the yeast YLY01; CG: blank control group treated sterilized seawater. Data are given as mean ± SD (n = 3).

(Fig 4). However, in the control group, the ammonia nitrogen concentration in SFW nearly remained unchanged (Fig 4). It demonstrated that the yeast YLY01 has a strong ability to remove ammonia nitrogen in water when little carbon source was added. During this process, no nitrite was detected in the experimental group or the control group.

## Inhibition of the yeast YLY01 to the growth of *Vibrio* spp.

Two methods were used to assay the inhibitory activity of the strain YLY01 against *Vibrio* spp.. The number of *Vibrio* cells in farming water of the control group was approximately $4.6 \times 10^4$ cfu/mL at the start (T0)and slightly decreased to $3.8 \times 10^4$ cfu/mL at 48 h, while the number of *Vibrio* cells in farming water of the experiment group (EG) decreased from $4.6 \times 10^4$ cfu/mL at the start to $6.8 \times 10^3$ cfu/mL at 48 hr, which was only 18% of the *Vibrio* cells in the control group (CG) at the same time point (Fig 5). This result demonstrated that the existence of the yeast YLY01 cells inhibited the survival of *Vibrio* cells in farming water.

To further explore what caused the inhibition to the growth of *Vibrio* spp., the plate suppression experiment of the strain YLY01 was also conducted. No inhibition zones were observed when the YLY01 cells were applied whilst inhibition zones in the positive control group (added with chloramphenicol) were clear and obvious (Fig 6). This result clearly indicated that the YLY01 cells could not generate antimicrobial substances against *Vibrio* cells. Therefore, the obvious decrease in numbers of *Vibrio* cells during incubating with the YLY01 cells in farming water was probably caused by other mechanisms.

## Assessment of biological safety on shrimp *Litopenaeus vannamei*

Under normal culture conditions, the survival rates of shrimps in different treatments were shown in the Fig 7. The result indicated that the survival rate of shrimps in the blank control group (C0, treated with sterilized seawater) slightly decreased to 86.7% after 7-day culture. The survival of rate of shrimps in the experimental group (EG, challenged by the strain YLY01) reach up to 96.7%, which was even higher than the survival rate of shrimp in the blank control group. The survival rate of shrimps in the positive control group (C1, challenged by *Vibrio harveyi*) gradually declined, and after seven days the survival rate of shrimps in the group was only 40.0%. Considering a big dosage of the strain YLY01 in the experimental group

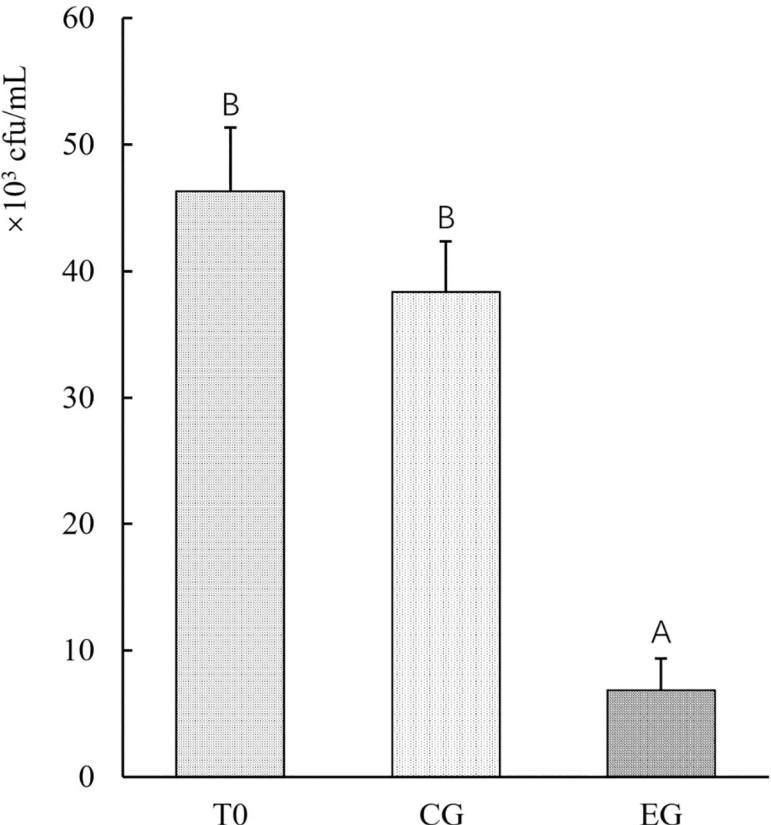

**Fig 5. The number of *Vibrio* cells in farming water after 48-h incubation.** T0: the start of experiment at 0 h; CG: the control group without the supplement of the yeast YLY01; EG: the experimental group supplemented by the yeast YLY01. Data are given as mean ± SD (n = 3). Different letters above bars indicate significant difference among treatments (p < 0.05).

(approximately $1 \times 10^5$ cfu/mL), the result indicated that the strain YLY01 not only has high biosafety for *L. vannamei* but also has a potential to reduce the death rate of the cultured shrimps. Notably, much clearer water was observed in the experimental group than in blank control group and in the positive control group after the seven-day experiment, and obvious flocculent precipitate generated on the bottom of tanks in the experimental group.

## Nutritional components of the yeast YLY01

The nutritional composition of the yeast YLY01 is shown in Table 1. The biomass of the yeast YLY01 could reach up to in 26.5 g/L after 72-h culture, and the free water content of the cells was about 78%. The protein content of the yeast accounted for 30.3% of dry cell weight, and 16 hydrolyzed amino acids were detected and compared with counterparts in a marine red yeast, *R. paludigenum* [27], and brewer's yeast, *Saccharomyces cerevisiae* [28] (Table 2). There was no obvious difference in terms of the contents of nine essential amino acids for crustaceans [29] among the yeast YLY01 (50.2%), a marine red yeast *R. paludigenum* (52.2%), and *S. cerevisiae* (49.2%). The yeast YLY01, *R. paludigenum*, and *S. cerevisiae* also had similar total contents of six flavor amino acids (accounting for 43.1%, 45.6%, and 43.5%, respectively). The profile of amino acid composition of the yeast YLY01 was also compared with these of *R. paludigenum* and *S. cerevisiae* and they were exhibited in a radar plot (S1 Fig). Similar shapes in the plot manifested that three kinds of yeasts had roughly similar variation trends in terms of relative

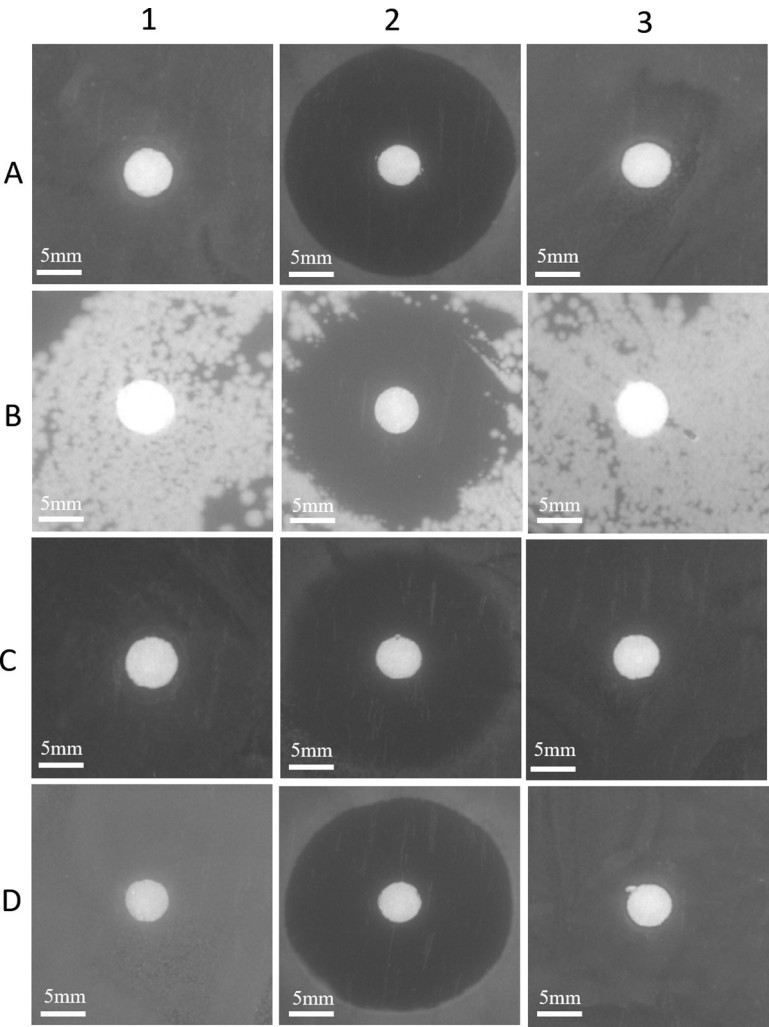

**Fig 6. Inhibition assay of the yeast YLY01against four *Vibrio* species.** A: *V. alginolyticus* E06333; B: *V. furnissi* ATCC 33813; C: *V. harveyi* E385; D: *V. parahaemolyticus* ATCC 17802; 1: experimental group (added with the YLY01 cell culture); 2: positive control (added with chloramphenicol); 3: negative control (added with the medium).

amino acid contents except the contents of histidine (His) and lysine (Lys), wherein the yeast YLY01 had almost twice the lysine (Lys) content of *R. paludigenum* and *S. cerevisiae*.

In addition, the fatty acids content was 3.24% of dry cell weight, and the unsaturated fatty acids could reach up to 85.2% of total fatty acids. The crude polysaccharides accounted for 16.8% of cell dry weight. Moreover, the carotenoid content was 1.25 mg/g. The yeast YLY01 was also detected to contain a small amount of B vitamins (0.15 mg/g).

## Discussion

Marine red yeasts are saprophytic organisms with a strong ability to resist adversity and they act as decomposer through converting plant and animal organics to yeast biomass in the natural environment [30], which enables them to have great potential to be explored as probiotics. So far as now, marine red yeasts have never been reported to clarify farming water and control *Vibrio* spp. in farming water though the ability of some marine red yeasts to utilize inorganic nitrogen has long been noticed or utilized by several researchers. Saenge et al. [31] revealed

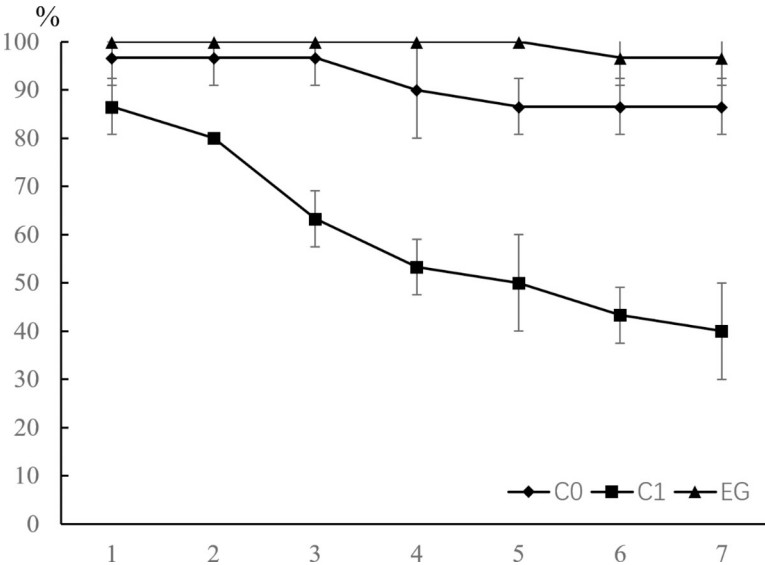

**Fig 7. Survival rates of shrimp *Litopenaeus vannamei* after the immersion challenge in seven days.** EG: the experimental group challenged by the yeast YLY01; C0: the blank control group; C1: the positive control group challenged by *Vibrio harveyi*. Data are given as mean ± SD (n = 3).

that the accumulation of lipids and carotenoids in *Rhodotorula glutinis* increased when ammonium sulfate was used as nitrogen source to change the C/N ratio in medium. Inorganic nitrogen is also used as a nitrogen source in some isolation media of *Rhodotorula* spp. [32, 33]. These findings implied that some marine red yeasts have the potential to remove ammonia nitrogen in farming water through ammonium assimilation, which is one important way to utilize inorganic nitrogen source by common yeast species such as *Saccharomyces cerevisiae* [34, 35], *Candida utilis* [34], and *Kluyveromyces marxianus* [36]. In this study, *R. sphaerocarpum* YLY01 was found to remove ammonia nitrogen with a high efficiency when carbon source was added and other nitrogen sources were limited, and meanwhile no nitrite was detected during this process. It indicated *R. sphaerocarpum* YLY01 can utilize ammonia nitrogen for growth through ammonium assimilation as other common yeast species do. Therefore, it is the first report that a marine red yeast can be used to remove ammonia nitrogen in farming water. In addition, we observed much clearer water and obvious flocculent precipitate in the experiment group of the biosafety assay, and it also indicated that *R. sphaerocarpum* YLY01 has an excellent flocculation ability. Yeast flocculation has been explored for long time [37] and the advantage of yeast flocculation has been taken to develop microbial flocculant for wastewater treatment [38]. It is worth to further explore the detailed flocculant profiles of *R. sphaerocarpum* YLY01 in future focusing on sewage (including aquaculture tail water) treatment.

In this study, *R. sphaerospora* YLY01 exhibited the strong inhibition ability to the survival of *Vibrio* cells in farming water when little carbon source was supplied, and the inhibition to *Vibrio* cells was not likely caused by generating antibiotic substances as no inhibition zones occurred. Therefore, *R. sphaerospora* YLY01 should have other antibacterial mechanisms.

**Table 1. The nutritional components of the yeast *R. sphaercarpum* YLY01.**

| Nutrition ingredients | Protein | Crude Polysaccharides | Saturated Fatty Acids | Unsaturated fatty acids | Carotenoid | B vitamins |
|---|---|---|---|---|---|---|
| Proportion in dry weight (%) | 30.3 | 16.8 | 0.48 | 2.76 | 0.125 | 0.015 |

**Table 2. The composition of hydrolyzed amino acids of the yeast *R. sphaercarpum* YLY01 and other two yeasts.**

| Groups (%) | Amino acids | Proportion (%) | | |
|---|---|---|---|---|
| | | **YLY01** | **1** | **2** |
| EAA | Phe* | 4.32 | 6.10 | 3.85 |
| | Met | 1.99 | 1.87 | 1.67 |
| | Arg | 7.89 | 8.45 | 6.92 |
| | Lys | 5.66 | 7.20 | 8.85 |
| | Leu | 8.24 | 8.95 | 7.69 |
| | Val | 5.96 | 6.64 | 6.28 |
| | Ile | 4.27 | 4.48 | 5.51 |
| | His | 6.60 | 3.53 | 3.21 |
| | Thr | 5.26 | 5.41 | 5.90 |
| NEAA | Ala* | 7.79 | 8.16 | 7.05 |
| | Pro | 5.16 | 2.83 | 5.13 |
| | Gly* | 5.56 | 6.14 | 5.13 |
| | Glu* | 12.41 | 11.62 | 13.97 |
| | Tyr* | 2.38 | 3.88 | 4.23 |
| | Ser | 5.86 | 5.03 | 5.38 |
| | Asp* | 10.67 | 9.70 | 9.23 |

EAA: essential amino acids; NEAA: nonessential amino acids; *: flavor amino acids; 1: *Rhodosporidium paludigenum* [27]; 2: *Saccharomyces cerevisiae* [28].

Anyway, this is the first report to confirm that a marine red yeast has inhibition ability to *Vibrio* cells in farming water, which will largely expand the usage of marine red yeasts.

The yeast *R. sphaerospora* YLY01 contains 16 hydrolyzed amino acids, which is consist with the kinds of amino acids found in *R. paludigenum* [27] and brewer's yeast *S. cerevisiae* [28]. The yeast YLY01 also had high total content of six flavor amino acids (up to 43.1%) compared with that of *R. paludigenum* [27] and brewer's yeast [28]. Researchers suggest that high content of flavor amino acids can greatly improve the palatability and food attractiveness of feed or live bacteria preparations [39]. The yeast YLY01 in this assay had a moderate lipid content (3.24%), and the ratio of unsaturated fatty acids (85.2%) was higher than that (approximately 74.2%) of the representative species, *R. toruloides*, in genus *Rhodosporidium* [40]. An obvious feature of members in genus *Rhodosporidium* is that they have prominent ability to accumulate lipids, which makes them to have a great potential in microbial lipid industry [41, 42], and thus we speculated that *R. sphaerospora* YLY01 can accumulate rich lipids under certain growing conditions. Unsaturated fatty acids play an important role in providing energy, forming high bioactive substances, regulating lipid metabolism, and immune function in aquatic animals [43]. Coyle et al. [44] found that unsaturated fatty acids supplied in the diet reduce the metabolic energy expenditure and improve growth rate or diet efficiency of largemouth bass. It is worthy to exploit the fatty acid composition and potential application value of lipids of the yeast YLY01 in future. The production capacity of carotenoids in the yeast YLY01 is 1.25 mg/g dry weight under the conventional cultivation condition. A high yield of carotenoids (35 mg/g dry weight) in *R. mucilaginosa* was recorded [45], it suggests that the yield of carotenoids in *R. sphaerocarpum* YLY01 can be largely improved under suitable conditions if needed. Besides, *R. sphaerocarpum* YLY01 contained 16.79% crude polysaccharides, which are considered to stimulate the immune system and antioxidant systems of cultured animals and thus to have positive effects on reducing the risk of disease outbreaks and improving the resistance to adverse circumstances [11, 46]. *Rhodosporidium* is one of common marine red yeasts

frequently isolated from various marine environments [30], however the yeasts in *Rhodosporidium* were exploited as probiotics for animals just in recent years. The yeast, *R. paludigenum*, enhanced the growth performance and antioxidant competence of *L. vannamei* when it was added into the diet in forms of dried yeast or live bait [47]. The dietary addition of *Rhodotorula* (*Rhodosporidium*) *benthica* D30 could increase growth performance and some digestive enzyme activities, improve immunity and disease resistance of sea cucumber *A. japonicus* [33]. These rigorous and informative evidences from congeneric yeast species and the assays on nutrition and biosecurity of *R. sphaerocarpum* YLY01 provide our impetus to further explore probiotic functions of *R. sphaerocarpum* YLY01 and exploit it as a promising aquatic probiotics.

In aquaculture, *Bacillus* spp., lactic acid bacteria, and nitrifying bacteria are most common probiotics. *Bacillus* spp. generally improve animal growth/ feed utilization due to high activities of extracellular enzymes screened by them [48]. Lactic acid bacteria are mainly used to adjust regulate intestinal flora balance and animal immunity [49]. Nitrifying bacteria (mainly refer to ammonia-oxidizing bacteria [AOB] and nitrite-oxidizing bacteria [NOB]) were adopted to transform toxic ammonia nitrogen or nitrite into nitrate, and the metabolites and the growth speed of AOB and NOB are much lower than heterotrophic bacteria [50, 51], which leads to low efficiency of ammonia or nitrite removal by them in farming ponds. Compared to these conventional probiotics that only have one prominent function, the red yeast YLY01 exhibited versatile functions including high-efficiency ammonia removal without its transformation to nitrite, clarifying water, inhibition on *Vibrio* spp., and rich nutrition, which confers the red yeast YLY01 a good prospect to be developed as aquatic probiotics. Besides, it also shed some light on exploring new applications of marine red yeasts.

## Conclusion

In summary, the yeast *R. sphaerocarpum* YLY01 exhibits multiple prominent advantages including efficiently removing ammonia nitrogen without transforming it to nitrite, clarifying farming water, the inhibition to *Vibrio* spp., and rich nutrition, which makes it has a great potential to be developed as a versatile aquatic probiotic and as an effective microbial agent for high-ammonia sewage treatment.

## Supporting information

**S1 Fig. The radar plot of amino acid composition of three yeasts.**
(DOCX)

## Acknowledgments

We thank the manager Huo Li from Maoming Jinyang Aquaculture Company for provide an experimental base and samples.

## Author Contributions

**Data curation:** Long Yun, Wei Wang, Yingying Li.

**Formal analysis:** Long Yun, Wei Wang, Mei Xie, Ting Chen, Daning Li.

**Funding acquisition:** Peng Luo.

**Investigation:** Long Yun, Wei Wang.

**Methodology:** Peng Luo.

**Project administration:** Chaoqun Hu.

**Supervision:** Chaoqun Hu, Peng Luo.

**Writing – original draft:** Long Yun, Wei Wang.

**Writing – review & editing:** Peng Luo.

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
