## [Decision Letter · Decision Letter 0]

6 Jan 2021

PONE-D-20-38047

Potential application values of a marine red yeast, Rhodosporidiums sphaerocarpum YLY01, in aquaculture and wastewater treatment assessed by the removal of ammonia nitrogen, the inhibition to Vibrio spp., and nutrient composition

PLOS ONE

Dear Dr. Luo,

Thank you for submitting your manuscript to PLOS ONE. After careful consideration, we feel that it has merit but does not fully meet PLOS ONE’s publication criteria as it currently stands. Therefore, we invite you to submit a revised version of the manuscript that addresses the points raised during the review process.

We look forward to receiving your revised manuscript.

Kind regards,

Yiguo Hong

Academic Editor

PLOS ONE

Journal Requirements:

2. In your Methods section, please provide additional information regarding the permits you obtained to collect samples for the present study. Please ensure you have included the full name of the authority that approved the field site access and, if no permits were required, a brief statement explaining why.

Reviewers' comments:

Reviewer's Responses to Questions

**Comments to the Author**

1. Is the manuscript technically sound, and do the data support the conclusions?

Reviewer #1: Yes

Reviewer #2: Yes

Reviewer #3: Yes

2. Has the statistical analysis been performed appropriately and rigorously? 

Reviewer #1: Yes

Reviewer #2: Yes

Reviewer #3: Yes

3. Have the authors made all data underlying the findings in their manuscript fully available?

Reviewer #1: Yes

Reviewer #2: Yes

Reviewer #3: Yes

4. Is the manuscript presented in an intelligible fashion and written in standard English?

Reviewer #1: Yes

Reviewer #2: Yes

Reviewer #3: Yes

5. Review Comments to the Author

Reviewer #1: This manuscript by Yun et al., isolated and purified a marine red yeast strain YLY01 from farming wastewater and identified it as a member of Rhodosporidiums sphaerocarpum by the phylogeny. They further found that the strain YLY01 has high biosecurity to at least shrimps, and it can inhibit the growth of Vibrio cells when a small quantity of carbon source was added into farming water. Therefore, they concluded that the yeast YLY01 has a great potential to be used as a versatile probiotic in aquaculture. The topic is interesting, technically sound, and they use some cutting-edge technologies. In my opinion, it could be accepted for publication in PLOS ONE after some revisions.

Here are some suggestions for the manuscript:

1. On line 63, “Healthy-looking shrimp juveniles (4 ± 1 cm), Litopenaeus vannamei, ……”, the weight and quantity of the shrimp should be described here.

2. On line 64-66, “shrimps were further randomly sampled for conventional PCR detection to exclude the infection by pandemic pathogens, including WSSV [12], EHP [13], EMS-related vibrios.” What are the specific methods?

3. On line 136, a bracket is missing.

4. Figures must show datapoints, or the number of biological and technical replicates must be shown in the figure text.

5. The figure legends should be more detailed.

6. The quality of the figures should be improved in general. In figure 1 and 5, authors should include the statistical analysis with p-values (not asterisks) between different groups.

Reviewer #2: Manuscript Review (PONE-D-20-38047) - Potential application values of a marine red yeast, Rhodosporidiums sphaerocarpum YLY01, in aquaculture and wastewater treatment assessed by the removal of ammonia nitrogen, the inhibition to Vibrio spp., and nutrient composition.

Comments

This is an interesting manuscript that reports a marine red yeast, Rhodosporidiums sphaerocarpum YLY01, which could be used as a potential versatile probiotic in aquaculture. In this study, the authors isolated and purified a marine red yeast strain YLY01 from farming wastewater, which is suggested as a member of Rhodosporidiums sphaerocarpum. Through various physiological experiments, the strain YLY01 was found with the abilities to remove ammonia nitrogen in the farming water, inhibit the growth of Vibrio cells and protect the shrimps in normal culture conditions. Based on these interesting findings, the authors suggest that the red yeast strain YLY01 could be developed as a potential aquatic probiotic as well as an effective microbial agent for high-ammonia sewage treatment.

The authors have presented a compelling story with well controlled experiments and clear presentation of the data. The data of the biosecurity of the strain YLY01 to the shrimps would be more complete if the survival rate of the shrimps is tested when incubated with Vibrio harveyi and the red yeast strain YLY01 together. Please find the questions and suggestions listed below.

Major questions and suggestions:

1. Line 183: The yeast YLY01 has a strong ability to remove ammonia nitrogen in water when little carbon source was added. What about the ammonia nitrogen removal ability with large amount of carbon source? Did the authors test it?

2. Line 185: Why the inhibition of the yeast YLY01 to the growth of Vibrio spp was observed in CFU but not in plate suppression experiment? Could the authors please provide some suppositions to explain it?

3. Line 199: In the assessment of biological safety, did the authors test the survival rate of the shrimps when incubated with Vibrio harveyi and the red yeast strain YLY01 together? Can the yeast YLY01 protect the shrimps in the present of Vibrio harveyi?

Minor suggestions:

1. Line 44: ‘Of those methods’ could be ‘Among those methods’.

2. Line 45: ‘Yeasts as one group of probiotics have been mainly used either as fresh baits…’ could be ‘Yeasts, as one group of probiotics, have been mainly used either as fresh baits…’.

3. Line 202: ‘in 7-day culture’ could be ‘after 7-day culture’.

4. Line 224: The authors could add the full name of ‘His and Lys’, namely, ‘Histidine (His) and Lysine (Lys)’.

Reviewer #3: The manuscript by Long Yue et al investigated Potential application values of a marine red yeast, Rhodosporidiums sphaerocarpum YLY01, in aquaculture and wastewater treatment assessed by the removal of ammonia nitrogen, the inhibition to Vibrio spp., and nutrient composition, which provided an useful strategy to treatment the high-ammonia tail water in aquaculture. Overall, the study was well designed and executed, date properly analyzed, and results well discussed, although some minor revision was needed.

1.It is better to change the wastewater into tail water. Please do throughout the manuscript.

The form of N in the aquatic water was decided by the water temperature and pH. However, I find the pH value was not introduced in ammonia nitrogen removal trial. Please add.

Line 136, add the other parenthesis.

Line 143, how much see water were complemented each day each tank?

Line 226, the table 1 does not seem pretty enough. It is better to change the items into the horizontal row.

Line 61, how long did you store the water samples at 4℃ before using, as we know, the species of microorganisms will slowly change even at 4℃ for long time and it is not the original sample.

Line 130, why do you select these four Vibrio strains as criterion to determine whether the strain can inhibit the growth of Vibrio species, are them more sensitive?

Line 165, it is better to change ‘‘of’’ to ‘‘then’’.

Line175-177, cite a reference about describing cell morphology of yeast.

Line 219, a percent sign missed when you describe S. cerevisiae.

6. PLOS authors have the option to publish the peer review history of their article (what does this mean?). If published, this will include your full peer review and any attached files.

Reviewer #1: No

Reviewer #2: No

Reviewer #3: No

---

## [Author Response · Author response to Decision Letter 0]

15 Jan 2021

Reviewer #1: 

This manuscript by Yun et al., isolated and purified a marine red yeast strain YLY01 from farming wastewater and identified it as a member of Rhodosporidiums sphaerocarpum by the phylogeny. They further found that the strain YLY01 has high biosecurity to at least shrimps, and it can inhibit the growth of Vibrio cells when a small quantity of carbon source was added into farming water. Therefore, they concluded that the yeast YLY01 has a great potential to be used as a versatile probiotic in aquaculture. The topic is interesting, technically sound, and they use some cutting-edge technologies. In my opinion, it could be accepted for publication in PLOS ONE after some revisions.

Here are some suggestions for the manuscript:

1. On line 63, “Healthy-looking shrimp juveniles (4 ± 1 cm), Litopenaeus vannamei, ……”, the weight and quantity of the shrimp should be described here.

R: Thank you for your suggestion. We added the weight and quantity of the shrimp here.

2. On line 64-66, “shrimps were further randomly sampled for conventional PCR detection to exclude the infection by pandemic pathogens, including WSSV [12], EHP [13], EMS-related vibrios.” What are the specific methods?

R: I am sorry. The sentence is ambiguous, so we rewrote it to avoid the misunderstanding. Actually, the detection methods are included in the references ([12], [13], and [14]). The three references were not adopted to introduce three types of pathogens. 

3. On line 136, a bracket is missing. 

R: We added it.

4. Figures must show datapoints, or the number of biological and technical replicates must be shown in the figure text. 

R: According to the comment 6, figure 1 and 5 should include the statistical analysis with p-values. We showed statistical analysis in new figure 1 and 5, and thus it is inaesthetic to show datapoints as well in the same figure. It is hard to show datapoints in a dynamic curve in figure 4 and 7, and also it is not the conventional method to mark datapoints in this kind of figure. Thank you for your understanding.

5. The figure legends should be more detailed.

R: We rewrote the figure legends and increased some details. And also the legends of figure 1, 4, 5, and 7 included the description of statistical analysis.

6. The quality of the figures should be improved in general. In figure 1 and 5, authors should include the statistical analysis with p-values (not asterisks) between different groups.

R: Thank you for your good suggestions. We have done it as you said in figure 1 and 5. Several boxes were added in figure 3 to show the bud marks of the yeast cells. And we also improve the style of figure 1, 5, and 7 to improve the quality of the figures. 

Reviewer #2: 

Manuscript Review (PONE-D-20-38047) - Potential application values of a marine red yeast, Rhodosporidiums sphaerocarpum YLY01, in aquaculture and wastewater treatment assessed by the removal of ammonia nitrogen, the inhibition to Vibrio spp., and nutrient composition.

Comments

 This is an interesting manuscript that reports a marine red yeast, Rhodosporidiums sphaerocarpum YLY01, which could be used as a potential versatile probiotic in aquaculture. In this study, the authors isolated and purified a marine red yeast strain YLY01 from farming wastewater, which is suggested as a member of Rhodosporidiums sphaerocarpum. Through various physiological experiments, the strain YLY01 was found with the abilities to remove ammonia nitrogen in the farming water, inhibit the growth of Vibrio cells and protect the shrimps in normal culture conditions. Based on these interesting findings, the authors suggest that the red yeast strain YLY01 could be developed as a potential aquatic probiotic as well as an effective microbial agent for high-ammonia sewage treatment.

 The authors have presented a compelling story with well controlled experiments and clear presentation of the data. The data of the biosecurity of the strain YLY01 to the shrimps would be more complete if the survival rate of the shrimps is tested when incubated with Vibrio harveyi and the red yeast strain YLY01 together. Please find the questions and suggestions listed below.

Major questions and suggestions:

1. Line 183: The yeast YLY01 has a strong ability to remove ammonia nitrogen in water when little carbon source was added. What about the ammonia nitrogen removal ability with large amount of carbon source? Did the authors test it? 

R: Thank you for your good thinking. Little carbon source was supplied in the test as we mainly considered that carbon source is unlikely to largely supplied in actual farming tailwater treatment due to the cost of treatment. Therefore, we didn’t test it, however it can be theoretically predicted that more carbon source will increase nitrogen removal under the condition of the appropriate carbon/nitrogen ratio. 

2. Line 185: Why the inhibition of the yeast YLY01 to the growth of Vibrio spp was observed in CFU but not in plate suppression experiment? Could the authors please provide some suppositions to explain it?

R: Plate suppression experiment can indicate whether the tested microbes can produce antibacterial substances through the occurrence of inhibition zones. In this study, plate suppression experiment clearly exhibited that the yeast YLY01 could not generated antibacterial substances against Vibrio spp. When the yeast YLY01 was added into the farming water supplied with small carbon source (glucose) and yeast extract, the survival of Vibrio spp. was inhibited. We considered that it was caused by the discrepant utilization ability to one certain carbon source by different microbes. As we know, sugars are the necessary ingredients in the yeast medium for yeast growth as yeast cells can quickly utilize various simple sugars (e.g. glucose and sucrose) as best carbon source while for most heterotrophic bacteria such as vibrios and Escherichia coli, simple sugars (e.g. glucose and sucrose) are not best carbon sources. In another hand, the utilization of simple sugars (e.g. glucose and sucrose) will result in the excess production of organic acid, which decrease the pH and inhibit the growth of heterotrophic bacteria (Suzuki et al, 2000). In our observation, the yeast YLY01 can grow well and quickly in a medium containing rich glucose or sucrose, and in this process pH in the medium descended to pH 5, but the yeast YLY01 could still grow well, which indicate that the yeast YLY01 is not sensitive to low pH.

Reference: Suzuki H, Kishimoto M, Kamoshita Y, et al. On-line control of feeding of medium components to attain high cell density. Bioprocess Engineering, 2000, 22: 433–440.

3. Line 199: In the assessment of biological safety, did the authors test the survival rate of the shrimps when incubated with Vibrio harveyi and the red yeast strain YLY01 together? Can the yeast YLY01 protect the shrimps in the present of Vibrio harveyi?

R: We test survival rate of the shrimps when incubated with Vibrio harveyi or the red yeast strain YLY01. This experiment was mainly designed to test the biological safety of the strain YLY01 to shrimp Litopenaeus vannamei, and Vibrio harveyi was used as a comparison to indicate that the strain YLY01 is not harmful. The protection effect of the yeast YLY01 against the infection by pathogenic Vibrio harveyi is another topic. Anyway, this is a good suggestion, and we will adopt it in future research.

Minor suggestions

1. Line 44: ‘Of those methods’ could be ‘Among those methods’.

R: We revised it.

2. Line 45: ‘Yeasts as one group of probiotics have been mainly used either as fresh baits…’ could be ‘Yeasts, as one group of probiotics, have been mainly used either as fresh baits…’.

R: We revised it.

3. Line 202: ‘in 7-day culture’ could be ‘after 7-day culture’.

R: We revised it.

4. Line 224: The authors could add the full name of ‘His and Lys’, namely, ‘Histidine (His) and Lysine (Lys)’.

R: We revised it.

Reviewer #3: 

The manuscript by Long Yue et al investigated Potential application values of a marine red yeast, Rhodosporidiums sphaerocarpum YLY01, in aquaculture and wastewater treatment assessed by the removal of ammonia nitrogen, the inhibition to Vibrio spp., and nutrient composition, which provided an useful strategy to treatment the high-ammonia tail water in aquaculture. Overall, the study was well designed and executed, date properly analyzed, and results well discussed, although some minor revision was needed.

1.It is better to change the wastewater into tail water. Please do throughout the manuscript. 

R: We changed “wastewater” into “tail water” at appropriate places.

2. The form of N in the aquatic water was decided by the water temperature and pH. However, I find the pH value was not introduced in ammonia nitrogen removal trial. Please add.

R: We added pH values of the medium and the simulated farming water.

3. Line 136, add the other parenthesis.

R: We added a missing bracket.

4. Line 143, how much see water were complemented each day each tank?

R: Approximately 0.5 L of sterilized seawater was complemented every day. We added this number at an appropriate place in the context. In order to avoid the effect of changing water on the number of tested yeast and Vibrio cells, we try to decrease the complement of sterilized seawater. To achieve this goal, residual feeds and feces were siphoned quickly as the manuscript described. 

5. Line 226, the table 1 does not seem pretty enough. It is better to change the items into the horizontal row.

R: We revised the format of table 1.

6. Line 61, how long did you store the water samples at 4℃ before using, as we know, the species of microorganisms will slowly change even at 4℃ for long time and it is not the original sample.

R: The collected water samples were stored at 4 ℃ in a refrigerator before using, but actually they were used quickly in 12 h. Our aim is to isolate yeasts that can remove ammonia nitrogen not to investigate the community of microbes, therefore the storing temperature of 4 ℃ has little effect on acquiring the interesting yeast strains. Actually, the storing temperature of 4℃ is suitable to remain the survival of microbes and slow down the change of microbial community.

7. Line 130, why do you select these four Vibrio strains as criterion to determine whether the strain can inhibit the growth of Vibrio species, are them more sensitive?

R: We chose these Vibrio species as they have some representatives. V. alginolyticus is a most common Vibrio species in various marine environments, and also some strains were reported to be opportunistic pathogen. Vibrio furnissi, V. harveyi, and V. parahaemolyticus are usual pathogenic Vibrio species for aquatic animals. Until now, there is no report on the discrepancy of sensitivity of Vibrio strains to specific antagonistic microbes. 

8. Line 165, it is better to change ‘‘of’’ to ‘‘then’’.

R: We changed it.

9. Line175-177, cite a reference about describing cell morphology of yeast.

R: we cite a new reference here to meet this requirement.

10. Line 219, a percent sign missed when you describe S. cerevisiae.

R: We added it.

---

## [Decision Letter · Decision Letter 1]

27 Jan 2021

Potential application values of a marine red yeast, Rhodosporidiums sphaerocarpum YLY01, in aquaculture and tail water treatment assessed by the removal of ammonia nitrogen, the inhibition to Vibrio spp., and nutrient composition

PONE-D-20-38047R1

Dear Dr. Luo,

We’re pleased to inform you that your manuscript has been judged scientifically suitable for publication and will be formally accepted for publication once it meets all outstanding technical requirements.

Kind regards,

Yiguo Hong

Academic Editor

PLOS ONE

Reviewers' comments:

Reviewer's Responses to Questions

**Comments to the Author**

1. If the authors have adequately addressed your comments raised in a previous round of review and you feel that this manuscript is now acceptable for publication, you may indicate that here to bypass the “Comments to the Author” section, enter your conflict of interest statement in the “Confidential to Editor” section, and submit your "Accept" recommendation.

Reviewer #1: All comments have been addressed

Reviewer #2: All comments have been addressed

2. Is the manuscript technically sound, and do the data support the conclusions?

Reviewer #1: Yes

Reviewer #2: Yes

3. Has the statistical analysis been performed appropriately and rigorously? 

Reviewer #1: Yes

Reviewer #2: Yes

4. Have the authors made all data underlying the findings in their manuscript fully available?

Reviewer #1: Yes

Reviewer #2: Yes

5. Is the manuscript presented in an intelligible fashion and written in standard English?

Reviewer #1: Yes

Reviewer #2: Yes

6. Review Comments to the Author

Reviewer #1: The authors have satisfactorily answered to the minor point suggested by this reviewer. I recommend the paper for publication.

Reviewer #2: In the revised manuscript, all the questions and comments have been well addressed, and it presented a compelling story with well controlled experiments and clear presentation of the data. Therefore, I suggest that this manuscript (PONE-D-20-38047R1) could be acceptable for publication in the PLOS ONE.

7. PLOS authors have the option to publish the peer review history of their article (what does this mean?). If published, this will include your full peer review and any attached files.

Reviewer #1: No

Reviewer #2: No

---

## [Editor Report · Acceptance letter]

3 Feb 2021

PONE-D-20-38047R1 

Potential application values of a marine red yeast, *Rhodosporidiums sphaerocarpum* YLY01, in aquaculture and tail water treatment assessed by the removal of ammonia nitrogen, the inhibition to *Vibrio* spp., and nutrient composition 

Dear Dr. Luo:

I'm pleased to inform you that your manuscript has been deemed suitable for publication in PLOS ONE. Congratulations! Your manuscript is now with our production department. 

Kind regards, 

on behalf of

Professor Yiguo Hong 

Academic Editor

PLOS ONE